# Has the COVID-19 Pandemic Changed Parental Attitudes and Beliefs Regarding Vaccinating Their Children against the Flu?

**DOI:** 10.3390/vaccines11101519

**Published:** 2023-09-24

**Authors:** Liora Shmueli

**Affiliations:** Department of Management, Bar-Ilan University, Ramat Gan 52900, Israel; liora.shmueli@biu.ac.il; Tel.: +972-50-223-6382

**Keywords:** health belief model, flu, SARS-CoV-2, vaccine acceptance

## Abstract

Background: This study assessed whether the COVID-19 pandemic has altered parents’ attitudes toward vaccinating their children against the flu and barriers to school-based vaccination programs. Methods: A cross-sectional online survey was conducted with 975 parents of children aged 6 months to 11 years between 21–31 December 2022. A multivariate regression was performed to determine predictors of parents’ willingness to vaccinate their children against the flu in the winter of 2023. Results: 45% of parents did not plan to vaccinate their children against the flu, citing concerns about side effects and vaccine effectiveness; 39% already vaccinated their children, and 41% of them reported an increased intention to vaccinate following the pandemic. Only 37% of parents chose school-based vaccination programs, mainly due to a preference for HMO clinics and a lack of available nurses at school. The Health Belief Model variables, namely, perceived susceptibility, severity, and benefits, displayed the largest effect sizes. Conclusions: Healthcare providers and public health officials should address parents’ concerns about flu vaccine safety and efficacy to improve vaccination rates among children. Notably, the pandemic has increased vaccine receptivity among some parents. Enhancing accessibility to nursing staff in student health facilities could help boost vaccine uptake.

## 1. Introduction

Seasonal influenza, a contagious, infectious viral disease that affects the respiratory system, is caused by the influenza virus, and is mainly common in the fall and winter months. The main symptoms are a runny nose, dry cough, sore throat, fever over 39 °C, headache, and muscle pain. The influenza virus may even cause complications such as secondary infection, in particular pneumonia, hospitalization, and even death. Previous studies have estimated that 99% of deaths in children under 5 years of age with lower respiratory tract infections associated with influenza were in low- and middle-income countries (LMIC) [1,2].

The most effective way and primary approach recommended by the World Health Organization (WHO) and the U.S. Centers for Disease Control and Prevention (CDC) to prevent influenza (flu) is that every person over the age of 6 months should get a flu vaccine every year at the beginning of the flu season. Flu shots are especially recommended for children between the ages of 6 months and 5 years, as they are a high-risk group [2,3].

In order to increase vaccination rates, since 2008, Israel has begun providing free influenza vaccinations for the entire population over the age of 6 months, and the Ministry of Health and the Health Maintenance Organizations (HMOs) have implemented various strategies to promote vaccination, especially among high-risk groups. Nonetheless, despite these efforts, studies conducted before the COVID-19 pandemic found a low degree of flu vaccine compliance in many countries across the globe, including Israel when it came to the vaccination of children. Specifically, in the years 2010–2017, only 51–59% of this age group got vaccinated in the U.S. [4,5], and in Israel, the numbers were even lower: 23–32% [6,7,8]. These figures are far below the Healthy People 2020 goal of 70% for children [9].

In the early stages of the COVID-19 pandemic, Israel emerged as a global leader in vaccination efforts, with remarkably high uptake rates of the COVID-19 vaccine [10], compared to efforts for the flu vaccine. Following an extensive COVID-19 vaccine campaign, it is intriguing to examine the factors that will impact parents on their decision to vaccinate their children against influenza following the pandemic.

Interestingly, several studies, including systematic reviews, demonstrated a mixed impact of the COVID-19 pandemic on parents’ intention to vaccinate their children against influenza during the pandemic. On the one hand, some studies reported that the pandemic had a positive impact, with parents being more likely to vaccinate their children against influenza, a shift attributed to changes in risk perception due to the COVID-19 pandemic and increased awareness of the importance of vaccination [11]. Similarly, Kong’s systematic review of the adult population during the COVID-19 pandemic found a heightened willingness to vaccinate against influenza worldwide [12]. On the other hand, other studies reported that the pandemic had a negative impact, with parents being less likely to vaccinate their children due to increased vaccine hesitancy during the COVID-19 pandemic [13,14].

Several factors were found to influence parents’ intention to vaccinate their children against influenza during the COVID-19 pandemic. These include parents’ perception of the severity and risk of COVID-19 and influenza, level of trust in healthcare providers and government recommendations, previous experience with influenza vaccination, and socio-demographic characteristics. Older age, Black race, and co-pay have all been associated with decreased influenza vaccine administration [13].

Only a few studies have investigated caregivers’ hesitancy toward influenza vaccination and assessed the associated factors during the COVID-19 pandemic using the Health Belief Model (HBM), e.g., [15]. The HBM was widely used in the context of vaccination, particularly influenza vaccination, prior to the COVID-19 pandemic, especially among adults [16,17,18]. This model proposes several factors that are associated with influenza vaccine acceptance: perceived disease severity; perceived susceptibility; perceived benefits; perceived barriers; and cues to action [19,20]. Previous studies conducted prior to the COVID-19 pandemic have shown that the most dominant factors influencing parents’ intentions to vaccinate based on the HBM are their perceived benefits, perceived susceptibility, and severity [21,22,23].

However, to the best of our knowledge, no studies have examined, using the HBM, the willingness of parents to vaccinate their children against influenza after experiencing the COVID-19 pandemic, during which period some children were vaccinated against COVID-19.

One way to increase flu vaccination rates in this target audience is to administer it in schools. In Israel, as of 2017, flu vaccines are administered in elementary schools through the Ministry of Health’s Student Health Services (in addition to healthcare maintenance organizations, HMOs), similar to the NHS school-aged immunization service in England. In this preventative, government-funded medical service, students in grades 2–4 are administered nasal spray vaccination by nurses during school hours. Yet, even though this service is provided throughout Israel, many children are not getting vaccinated against the virus. In 2021, over 50% of parents in Israel refused to vaccinate their children in these grades at school [24,25].

This paper addresses two unmet needs. The first is to assess parental attitude changes to flu vaccinations for their children as a consequence of the COVID-19 pandemic. The second is to identify contributing factors to parental decision-making with regard to vaccinating their children against the flu, in general, including socio-demographic, health-related, and behavioral factors based on the HBM, and with regard to administering flu vaccinations in school, in particular.

## 2. Methods

### 2.1. Study Design and Participants

We conducted a cross-sectional online survey of 975 Israeli parents to children between the ages of 6 months and 11 years from 21–31 December 2022. The survey was distributed by Sarid Research Institute for Research Services via an online panel containing a wide pool of potential interviewees who had consented to participate in surveys from time to time. Eligible participants were required to be (1) 18+ years old and (2) parents of a child aged 6 months to 11 years. To compose a representative sample of the Jewish adult population in Israel, respondents were sampled by layers based on age, gender, geographic area, and level of religiosity. Before distributing the questionnaire to all of the respondents, a pre-test was conducted on the first 100 respondents to examine the reliability of the questionnaire. The reliability was verified using a Cronbach α internal reliability test (the HBM section of the questionnaire obtained an internal consistency of Cronbach α = 0.85). These 100 respondents were also included in the larger study reported in this manuscript.

### 2.2. Ethical Considerations

This study was approved by the Ethics Committee for Non-clinical Studies of Bar Ilan University.

### 2.3. Questionnaire

The following sections describe the dependent and independent variables in the questionnaire and their operationalization in this study. The questionnaire consisted of 34 mandatory questions, divided into the following sections: (1) socio-demographic predictor variables; (2) health-related predictor variables; (3) HBM predictor variables; and (4) issues related to public health policy (e.g., getting vaccinated at school).

The parameters comprising the study measurements used to build the conceptual model are described in Appendix A.

### 2.4. Measurement and Variables

The first dependent variable was parents’ willingness to vaccinate their children aged 6 months to 11 years against the flu in the winter of 2023 (December 2020–February 2023), measured in 3 categories: (1) Yes, I have already vaccinated my child; (2) I intend to vaccinate my child in the current winter, and (3) No, I have not vaccinated my child and do not intend to.

The independent variables were grouped into three blocks:(1)Socio-demographic predictor variables: (1) age group; (2) gender; (3) level of education; (4) marital status; (5) socio-economic level (the socioeconomic level according to the Central Bureau of Statistics. It is based on the average gross income for a family in Israel, which is about ILS 19,300 per month when both spouses are employed and about ILS 9000 per month for a single person, grouped into three levels: low, medium, and high.); (6) periphery level, determined by the residential area (the “periphery” level according to the Israeli Central Bureau of Statistics. It is based on a peripheral index that combines two components: potential accessibility index of the local authority and proximity of the local authority to the boundary of the Tel Aviv district (www.cbs.gov.il). The peripheral index includes local authorities that were classified into 10 clusters. In this study, these clusters were grouped based on their periphery distribution scale into three groups: peripheral, intermediate, and central.); (7) religiosity level (secular, traditional, religious, orthodox); and (8) working as medical staff.(2)Health-related predictor variables: (1) whether the respondent got vaccinated against COVID-19; (2) whether the respondent already got vaccinated against flu this winter (i.e., around December 2022); (3) whether the children of the respondent got vaccinated against COVID-19; (4) whether a family member suffers from a chronic disease (one or more of the following: heart disease, vascular disease and/or stroke, diabetes mellitus, hypertension, chronic lung disease, including asthma or immune suppression); (5) the existence of past episodes of flu in the current winter; (6) the existence of past episodes of hospitalization in the family in the current winter.(3)HBM predictor variables: (1) perceived susceptibility (included two items); (2) perceived severity (included two items); (3) perceived benefits (included three items); (4) perceived barriers (included one item); (5) cues to action (included three items); and (6) attitude (included one item). The HBM items were measured on a 1–6 scale (1—strongly disagree; 6—strongly agree). Negative items were reverse-scored. Scores for each item were averaged to generate each of the HBM-independent categories. The Cronbach α internal reliability method revealed the internal consistency of the HBM section to be Cronbach α = 0.86 (Appendix A).

In addition, several issues related to public health policy were examined: (1) public trust in pharmaceutical companies and the Ministry of Health following the COVID-19 pandemic; (2) the main reasons for the reluctancy to vaccinate children against the flu in the winter of 2023; and (3) the main reasons for preferring to vaccinate children in grades 2–4 against flu at the HMO clinic instead of at school (both options are free of charge in Israel).

### 2.5. Statistical Analyses

Data from the online questionnaires were analyzed using SPSS 28 software. To test the reliability of the HBM measures, a Cronbach’s α test was used. To describe the characteristics of the study population, the following methods of descriptive statistics were employed: frequencies, percentages, averages, and standard deviations.

Relationships between dependent and independent variables were examined by univariate analysis. To test the relationship between the demographic variables and willingness to receive a flu vaccine, a series of χ^2^ tests were used. To test the mean differences in the HBM variables at different levels of flu vaccine willingness, we used analysis of variance (ANOVA) tests. Eta squared was used to measure the effect size for significant effects.

To test the multivariate effect of all the significant demographic and health-related variables along with the effects of the HBM variables on willingness to get a flu shot, a multinomial logistic regression analysis was conducted (multinomial logistic regression is an appropriate statistical method due to the categorical nature of the dependent variable, “willingness to vaccinate” measured in 3 distinct and non-ordinal categories: (1) Yes, I have already vaccinated my child; (2) I intend to vaccinate my child in the current winter, and (3) No, I have not vaccinated my child and do not intend to). Only the socio-demographic and health-related variables found to be significant (*p* < 0.05) in the univariate analysis were inserted into the regression model as predictors. All the HBM variables were entered into the model as predictors.

## 3. Results

### 3.1. Participant Characteristics

Overall, 975 (an invitation to fill out the questionnaire was sent to an online panel of 7200 participants with the goal of obtaining ~1000 filled questionnaires. The invitation to fill out the questionnaire expired after reaching 975 valid answers) parents of children aged 6 months to 11 years completed the survey; 52% of them were female (n = 505). Almost two-thirds of the parents (n = 617) were 18–39 years old. The participants were distributed nearly equally among the three socio-economic categories (low, medium, and high), with 16% of respondents (n = 156) living in geographically peripheral regions. A total of 40% of the respondents (n = 417) were secular, 59% (n = 578) held an academic degree, most (89%, n = 869) were married, and 25% (n = 239) stated having a family member with a chronic disease. The descriptive characteristics of the respondents are provided in Table 1.

The distribution of the sample closely matches that of the adult Jewish population in Israel (Appendix A). This is unsurprising as the sample was drawn in layers to satisfy certain characteristics (e.g., age, gender, level of religiosity, and geographical area). For example, approximately 60% of both the sample and the entire population were between the ages of 18 and 39.

### 3.2. Intention to Vaccinate Children Aged 6 Months to 11 Years against the Flu

When parents were asked about their willingness to vaccinate their children aged 6 months to 11 years against the flu in the winter of 2023, 45% (n  =  435) stated they had not vaccinated and did not intend to vaccinate their children; 39% (n= 378) responded that they had already vaccinated their children; and the remainder, 16% (n = 160), responded that they intend to vaccinate their children. Among those whose children had already been vaccinated or intend to vaccinate, 41% stated that their intention increased following the COVID-19 pandemic.

### 3.3. Univariate Analysis: Vaccinate Children Aged 6 Months to 11 Years against Flu

The results of the univariate analyses between the socio-demographic and health-related variables of parents and their willingness to vaccinate their children aged 6 months to 11 years against the flu are reported in Table 1.

The following socio-demographic variables were found to have a statistically significant correlation (*p* < 0.05) with willingness to vaccinate: age, gender, educational level, medical staff, and religiosity level. The health-related variables found to have a statistically significant correlation (*p* < 0.05) with willingness to vaccinate were: children were vaccinated in the previous year against COVID-19, parent was vaccinated against COVID-19 and/or flu in the current winter.

Specifically, the findings show that women were less likely to vaccinate their offspring compared to men (49% vs. 40%, respectively, χ^2^ (2) = 10.29, *p* = 0.006), as were parents under the age of 40 compared to older ones (47% vs. 41%, respectively, χ^2^ (2) = 12.49, *p* = 0.002), non-academics compared to those with an academic degree (55% vs. 40%, respectively, χ^2^ (2) = 15.46, *p* = 0.02), non-medical occupations compared to medical staff (46% vs. 30%, respectively, χ^2^ (2) = 6.41, *p* = 0.04), ultra-orthodox Jews (haredi) compared to secular Jews (60% vs. 39%, respectively, χ^2^ (6) = 26.31, *p* < 0.001).

Interestingly, parents whose children did not receive a COVID-19 vaccine in the year 2022 expressed significantly higher unwillingness to vaccinate their children against flu in the winter of 2023 than those whose children did receive a COVID-19 vaccine in the year 2022 (52% vs. 26%, respectively, χ^2^ (2) = 79.3, *p* < 0.001).

Parents who did not vaccinate against COVID-19 were more unwilling to vaccinate their children aged 6 months to 11 years against the flu compared to those who were vaccinated (57% vs. 38%, respectively, χ^2^ (4) = 58.17, *p* < 0.001). Lastly, parents who did not opt to get vaccinated against the flu in the current winter conveyed greater unwillingness to vaccinate their children aged 6 months to 11 years against the flu compared to those who were vaccinated (74% vs. 4%, respectively, χ^2^ (4) = 539.36, *p* < 0.001).

No significant difference was found for the other demographic variables assessed, i.e., socio-economic level, marital status, periphery level, a family member with a chronic disease, past parental episodes of flu in the current winter, and past episodes of hospitalization in the family in the previous year.

The results of the analysis of variance (ANOVA) aimed at testing the mean differences between the HBM variables at the different levels of readiness to vaccinate children aged 6 months to 11 years against flu in the winter of 2023 are reported in Table 2. The perceived susceptibility (F(2,970) = 361.7, *p* < 0.001, η^2^ = 0.43), perceived severity (F(2,970) = 15.47, *p* < 0.001, η^2^ = 0.03) perceived benefits (F(2,970) = 393.47, *p* < 0.001, η^2^ = 0.45) perceived barriers (F(2,970) = 94.9, *p* < 0.001, η^2^ = 0.16), cues to action (F(2,970) = 281.29, *p* < 0.001, η^2^ = 0.37), and attitude (F(2,970) = 53.43, *p* < 0.001, η^2^ = 0.10) all had a statistically significant effect on parents’ willingness to vaccinate their offspring.

### 3.4. Multivariate Analysis

We performed a multinomial logistic regression analysis of the multivariate effect of all the significant demographic and health-related variables along with those of the HBM variables on the willingness of parents to vaccinate their children aged 6 months to 11 years against the flu. The regression analysis accounted for an estimated 71% of the explained variance in the willingness to vaccinate children aged 6 months to 11 years against the flu in the winter of 2023 (Nagelkerke Pseudo R2 = 0.71) (see full regression coefficients in Table 3).

While none of the socio-demographic variables were significantly associated with the willingness to vaccinate children aged 6 months to 11 years against the flu, four health-related variables were found to be significant predictors. Specifically, an examination of the regression coefficients indicated that when all other variables are controlled, parents who vaccinated their children against COVID-19 in the previous year were 2.52 times more likely to vaccinate their children against the flu in the winter of 2023 than those who did not vaccinate their children against COVID-19 in the previous year (OR = 2.52, 95% CI. 1.28–4.95, *p* = 0.01).

Yet, parents who themselves got vaccinated against COVID-19 were 0.57 times less likely to vaccinate their children for flu in comparison to parents who did not opt for a COVID-19 vaccine (OR = 0.57, 95% CI. 0.35–0.93, *p* = 0.03). In the case of parents who got a flu shot in the winter of 2023, they were 24.37 times more likely to vaccinate their children against the flu and 6.25 times more likely to intend to vaccinate their children against the flu in comparison to those who did not get a flu shot that winter (OR = 24.37, 95% CI. 8.28–71.74, *p* < 0.001; OR = 6.25, 95% CI. 1.92–20.29, *p* < 0.001). Parents who intended to get a flu shot were 9.3 more likely to vaccinate their children against the flu and 16.46 times more likely to intend to vaccinate their children against the flu in the same winter in comparison to those who did not get a flu shot (OR = 9.3, 95% CI. 4.31–20.07, *p* < 0.001, OR = 16.46, 95% CI. 7.59–35.69, *p* < 0.001).

All the HBM tested variables were found to be significant predictors of the willingness of parents to vaccinate their children aged 6 months to 11 years against the flu. Specifically, for each unit increase in susceptibility, the odds of vaccinating increased by 1.88-fold, similar to the odds of intending to vaccinate by 1.89-fold (*p* < 0.001). For each unit increase in severity, the odds of vaccinating increased by 1.82-fold (*p* < 0.001). For each unit increase in benefits, the odds of vaccinating increased by 2.65-fold, and the odds of intending to vaccinate by 1.86-fold (*p* < 0.001). For each unit increase in barriers, the odds of vaccinating decreased by 0.75-fold, and the odds of intending to vaccinate by 0.74-fold (*p* = 0.01). For each unit increase in cues to action, the odds of vaccinating increased by 1.45-fold and the odds of intending to vaccinate by 1.37-fold (*p* = 0.01). A complete description of the model, goodness of fit indices, and regression coefficients are presented in Table 3.

### 3.5. Main Reasons for the Reluctancy to Vaccinate Children against the Flu in the Winter of 2023

Forty-five percent (n = 435) of the parents did not vaccinate and did not intend to vaccinate their children against the flu in the winter of 2023. The most common reasons for this decision were fear of side effects of the vaccine (28%) and concerns about vaccine effectiveness (22%). Other reasons were a lack of trust in the Ministry of Health (13%) or in pharmaceutical companies and vaccines (13%); of these 169 parents, 132 (78%) stated that the lack of trust increased following the COVID-19 events. Another stated reason was the impression that natural vaccination affords greater protection than an injected vaccine (17%) (see Appendix A for the complete list of reasons).

### 3.6. The Main Reasons for Preferring to Vaccinate Children in Grades 2–4 against the Flu at an HMO Instead of at the School

The main reasons that parents opted to vaccinate their children in grades 2–4 at an HMO clinic rather than through the Student Health Services at the elementary school during school hours with a nasal spray (n = 188) were: the preference to vaccinate in the presence of the parent (36%), no nurse availability, so there were no vaccination campaign at the school (23%), wanted to vaccinate the children at school, but missed the vaccination day at school (16%), and considering the nasal spray to be not effective, whereas the HMO clinics administered flu shots, perceived by the parent to be more effective (12%).

## 4. Discussion

In this study, we examined parents’ willingness to vaccinate their children aged 6 months to 11 years against the flu following the COVID-19 pandemic, in the winter of 2023, and to identify contributing factors to parental decision-making in this regard, including socio-demographic, health-related, and behavioral factors. In addition, we explored parents’ attitudes toward vaccinating their children in school settings as part of Student Health Services.

Previous research has examined parents’ attitudes toward vaccinating their children against the flu, although only a few studies have explored this in the context of the COVID-19 pandemic. The observed reluctance among almost half of the surveyed parents to vaccinate their children against the flu in the upcoming winter underscores a pressing public health concern. Our results align with a recent study conducted at the end of 2022 in 14 Eastern Mediterranean region (EMR) countries, which similarly demonstrated a high level of hesitancy among parents towards influenza vaccination for their children [26]. It is also in line with a study conducted among Chinese middle-school students, which found an overall vaccination rate of 38.2% in this group [27]. This correspondence across different regions underscores the global nature of this issue and suggests that similar factors may be at play in shaping parental attitudes toward childhood flu vaccination.

The socio-demographic characteristics we found to be associated with parents’ willingness to vaccinate their children against the flu align with a growing body of evidence in the field. These factors include male gender [28], older parents [29], higher levels of education [29], and a medical profession [30]. These findings underscore the need to consider socio-demographics when designing targeted interventions and campaigns to improve vaccination rates in specific subpopulations.

Our study has illuminated the significance of HBM variables, particularly perceived susceptibility, perceived severity, and perceived benefits, which displayed the largest effect sizes in the context of vaccine acceptance. Our findings, which are in line with previous studies in the context of flu vaccination [21,22,23], underscore the critical role of these constructs as predictors of vaccine acceptance and their utility for interventions designed to promote health behaviors. To formulate effective interventions, it is important to take into account previous successful strategies. Prior research has suggested that increasing parental perception of benefits (such as reducing the risk of infection or protecting those at high risk), severity, and perceived susceptibility may be effective strategies for public health interventions. Therefore, directing public health messages toward parents that may increase perceptions of susceptibility and severity of influenza, and perceived benefits of the vaccine may yield more favorable outcomes in terms of increasing vaccine uptake [22].

Understanding the impact of the COVID-19 pandemic on parents’ willingness to vaccinate their children against the flu post-pandemic is crucial. As our findings indicate, the pandemic has led to greater vaccine receptivity among some parents. Notably, approximately 41% of parents whose children have already been vaccinated or are planning to get vaccinated reported an increase in their intention to vaccinate against the flu following the pandemic. These findings are consistent with a systematic review that highlighted a rise in the intention to vaccinate against the flu during the pandemic [12]. Thus, the pandemic has provided a window of opportunity to promote influenza vaccination and decrease vaccine hesitancy among parents. The COVID-19 pandemic increased public awareness regarding the critical role vaccines play as the ultimate strategy for preventing infections and improving protection. It highlighted both human vulnerabilities to emerging infectious diseases, and the limitations of other disease control means (e.g., lockdowns). This awareness is likely to reflect improved vaccine uptake in other diseases, including influenza [31].

While the pandemic has increased awareness of the importance of vaccination, it has also heightened vaccine hesitancy among some parents, particularly those skeptical about vaccines in general. Most of the parental concerns raised in this study center around vaccine safety and efficacy issues. As has been previously reported, vaccine safety and potential side effects are the main barriers to vaccination [27,32].

Interestingly, the lack of trust in the Ministry of Health and in the pharmaceutical companies increased among 78% of those unwilling to vaccinate their children in our study following the COVID-19 events. This lack of trust can arise from a range of factors, such as misinformation, lack of transparency, and conspiracy theories about vaccines, all of which contribute to confusion and mistrust among some parents. Israel played a pioneering role as the first nation to embark on mass COVID-19 vaccination efforts. Prior to the release of early Israeli studies on the real-world effects of mass vaccination, considerable uncertainty prevailed regarding the vaccine’s effectiveness beyond the controlled environment of clinical trials. Another factor contributing to this mistrust was the non-disclosure of certain details regarding the purchase agreement with Pfizer at the time [33]. It is, therefore, important that healthcare providers and public health officials address these concerns, provide accurate information about the safety and efficacy of the influenza vaccine, and work towards rebuilding trust in the Ministry of Health to help increase the vaccination rate among parents and children.

Our results further indicate that only 37% of parents to kids in grades 2–4 chose to vaccinate their children at school, mainly because they preferred doing so at the HMO clinic in their presence or due to the unavailability of a nurse to administer the vaccine at the school. The lack of personnel at the school may lead parents to opt for alternative vaccination options, such as visiting the HMO clinic. Overall, while implementing school-based vaccination programs may be challenging, it is an important strategy for improving vaccination rates and promoting public health.

In Israel, influenza vaccination is not mandatory for school attendance, resulting in lower coverage rates compared to routine childhood vaccination. To increase vaccination uptake, several actions can be taken. First, nursing staff in student health centers should be made more accessible. Second, clearer explanations should be provided about the effectiveness of nasal spray vaccinations. Finally, more efforts should be invested in simple outreach activities [8]. For example, recent research has shown that sending behaviorally informed invitation letters and reminders can increase childhood influenza vaccinations through schools and general practitioners (GP) [34].

### Study Limitations

This study has several limitations that should be recognized when interpreting the results reported here. First, it is important to be aware of the predictive limitation of a cross-sectional study. Namely, since the exposure and outcome are simultaneously assessed, it is not possible to determine a temporal or causal relationship between them. Therefore, it cannot be ruled out that other variables created pseudo-correlations. Another limitation of this study is the lack of information regarding the panel participants who received invitations but did not participate in the survey. However, it should be noted that the panel participants who did respond constitute a representative sample. Moreover, the study used the self-report of influenza vaccine acceptance and intention to influenza vaccine in the coming winter, but the self-report of actual behavior may be biased, unlike monitoring actual vaccination. Finally, the study used a cross-sectional observational design that does not allow one to derive any causal conclusions.

## 5. Conclusions

Understanding the impact of the COVID-19 pandemic on parents’ willingness to vaccinate their children against the flu is crucial. Notably, the pandemic has led to enhanced vaccine receptivity among some parents. Healthcare providers and public health officials should address parents’ concerns about the safety and efficacy of the influenza vaccine as a means to improve vaccination rates among children. Implementing school-based vaccination programs is an important strategy for promoting public health, but may be challenging. To increase uptake, nursing staff in student health facilities should be more accessible, and clear explanations about the efficacy of nasal spray vaccinations should be provided.

## Figures and Tables

**Table 1 vaccines-11-01519-t001:** Characteristics of respondents by the intention to vaccinate their children against flu (n = 975).

	All Subjects		No Vaccination n = 435 (45%)		Vaccination n = 378 (39%)		Intent to Vaccinate n = 160 (16%)		χ^2^	*p*-Value
Sociodemographic	N	%	N	%	N	%	N	%		
Age group									12.49	0.002
18–39	617	63.3%	290	47.1%	214	34.7%	112	18.2%		
40–60	358	36.7%	145	40.6%	164	45.9%	48	13.4%		
Gender									10.29	0.006
Male	467	48%	186	39.8%	205	43.9%	76	16.3%		
Female	505	52%	247	49.1%	173	34.4%	83	16.5%		
Education level									15.46	0.02
High school or less	173	17.7%	81	47.1%	61	35.5%	30	17.4%		
Non-academic	224	23%	123	54.9%	72	32.1%	29	12.9%		
BA	418	42.9%	167	40%	177	42.4%	73	17.5%		
MA or higher	160	16.4%	64	40%	68	42.5%	28	17.5%		
Marital status									8.94	0.18
Single	54	5.5%	17	31.5%	29	53.7%	8	14.8%		
Married	869	89.1%	392	45.2%	327	37.7%	148	17.1%		
Divorced	48	4.9%	24	50%	20	41.7%	4	8.3%		
Widow	4	0.4%	2	50%	2	50%	0	0%		
Socio-economic level									7.37	0.12
Low	367	37.6%	180	49.3%	130	35.6%	55	15.1%		
Medium	303	31.1%	131	43.2%	115	38%	57	18.8%		
High	305	31.3%	124	40.7%	133	43.6%	48	15.7%		
Peripheral level									2.63	0.62
Periphery	156	16%	71	45.5%	65	41.7%	20	12.8%		
Intermediate	399	40.9%	183	46%	150	37.7%	65	16.3%		
Central	420	43.1%	181	43.2%	163	38.9%	75	17.9%		
Medical staff									6.41	0.04
No	912	93.5%	416	45.7%	345	37.9%	149	16.4%		
Yes	63	6.5%	19	30.2%	33	52.4%	11	17.5%		
Religiosity									26.31	<0.001
Secular	417	42.8%	163	39.2%	190	45.7%	63	15.1%		
Traditional	247	25.3%	117	47.6%	86	35%	43	17.5%		
Religious	128	13.1%	66	51.6%	36	28.1%	26	20.3%		
Haredi	101	10.4%	61	60.4%	26	25.7%	14	13.9%		
Health-related variables										
Chronic disease									3.6	0.17
No chronic disease	736	75.5%	339	46.2%	282	38.4%	113	15.4%		
Chronic disease	239	24.5%	96	40.2%	96	40.2%	47	19.7%		
Past episodes of flu current winter parent									3.57	0.17
No	803	82.4%	352	43.9%	310	38.7%	140	17.5%		
Yes	172	17.6%	83	48.5%	68	39.8%	20	11.7%		
COVID-19 vaccine last year—kids									79.3	<0.001
No	710	73.8%	365	51.6%	217	30.6%	126	17.8%		
Yes	252	26.2%	65	25.8%	157	62.3%	30	11.9%		
COVID-19 vaccine—parent									58.17	<0.001
No	390	41.2%	222	57.1%	105	27%	62	15.9%		
Yes	497	52.5%	186	37.5%	232	46.8%	78	15.7%		
Not yet, but intend to	60	6.3%	11	18.3%	34	56.7%	15	25%		
Past episodes of hospitalization									8.31	0.22
No	872	90.6%	399	45.9%	328	37.7%	143	16.4%		
Yes, due to flu complications	4	0.4%	0	0%	4	100%	0	0%		
Yes, due to COVID-19 complications	16	1.7%	6	37.5%	8	50%	2	12.5%		
Yes, other reasons	70	7.3%	28	40%	29	41.4%	13	18.6%		
Flu vaccine current winter—parent									539.36	<0.001
No	558	57.4%	410	73.6%	95	17.1%	52	9.3%		
Yes	224	23%	9	4%	189	84.8%	25	11.2%		
Not yet, but intend to	190	19.5%	16	8.4%	92	48.4%	82	43.2%		

**Table 2 vaccines-11-01519-t002:** Analysis of variance.

	No Vaccination		Vaccination		Intent to Vaccinate		F-Test	*p* Value (Two-Tail)	Effect Size (η^2^)
Variables	Mean	SD	Mean	SD	Mean	SD			
Susceptibility	2.50	1.13	4.49	1.12	4.16	0.94	361.70	<0.001	0.43
Severity	2.42	1.12	2.77	0.95	2.83	0.92	15.47	<0.001	0.03
Benefits	2.57	1.03	4.52	1.03	4.15	1.00	393.47	<0.001	0.45
Barriers	4.58	1.33	3.31	1.35	3.68	1.32	94.9	<0.001	0.16
Cues to action	2.37	1.18	4.18	1.12	4.00	1.11	281.29	<0.001	0.37
Attitude	4.03	1.65	4.97	1.20	4.95	1.08	53.43	<0.001	0.10

**Table 3 vaccines-11-01519-t003:** Multinomial logistic regression concerning the willingness to vaccinate children against the flu (n = 975).

	Vaccination						Intent to Vaccinate					
	B	SE	P	OR	LLCI	ULCI	B	SE	P	OR	LLCI	ULCI
Intercept	−7.57	1.21	0.00				−7.04	1.25	0.00			
Age Group (40+)	0.23	0.30	0.45	1.25	0.70	2.26	0.05	0.32	0.86	1.06	0.57	1.96
Sex (Male)	−0.17	0.27	0.53	0.84	0.49	1.44	0.14	0.29	0.62	1.16	0.66	2.03
Education (non-academic)	−0.22	0.43	0.61	0.80	0.34	1.86	−0.59	0.45	0.19	0.55	0.23	1.34
Education (BA)	0.06	0.40	0.87	1.07	0.49	2.32	−0.13	0.40	0.74	0.87	0.40	1.92
Education (MA or higher)	−0.33	0.47	0.48	0.72	0.29	1.80	−0.25	0.48	0.60	0.78	0.31	1.98
Medical staff	−0.25	0.61	0.68	0.78	0.24	2.56	−0.34	0.63	0.58	0.71	0.21	2.43
Religiosity (Traditional)	−0.02	0.32	0.94	0.98	0.52	1.85	−0.06	0.35	0.86	0.94	0.48	1.85
Religiosity (Religious)	−0.65	0.39	0.09	0.52	0.24	1.11	−0.11	0.39	0.78	0.90	0.42	1.94
Religiosity (Haredi)	−0.18	0.46	0.70	0.84	0.34	2.07	0.02	0.49	0.96	1.02	0.40	2.65
COVID-19 vaccine last year—kids	0.92	0.34	0.01	2.52	1.28	4.95	0.16	0.38	0.67	1.17	0.56	2.46
COVID-19 vaccine last year—parent	−0.46	0.24	0.05	0.63	0.40	1.00	−0.55	0.25	0.03	0.57	0.35	0.93
Current flu vaccine—parent	3.19	0.55	<0.001	24.37	8.28	71.74	1.83	0.60	<0.001	6.25	1.92	20.29
Positive intention flu vaccine—parent	2.23	0.39	<0.001	9.30	4.31	20.07	2.80	0.39	<0.001	16.46	7.59	35.69
Health Belief Model												
Susceptibility	0.63	0.18	<0.001	1.88	1.33	2.67	0.63	0.19	<0.001	1.89	1.31	2.71
Severity	0.60	0.16	<0.001	1.82	1.33	2.49	0.58	0.16	<0.001	1.79	1.30	2.47
Benefits	0.98	0.19	<0.001	2.65	1.82	3.87	0.62	0.20	<0.001	1.86	1.27	2.73
Barriers	−0.29	0.11	0.01	0.75	0.61	0.92	−0.30	0.11	0.01	0.74	0.59	0.92
Cues to action	0.37	0.14	0.01	1.45	1.10	1.91	0.32	0.15	0.03	1.37	1.03	1.83
Attitude	−0.15	0.12	0.21	0.86	0.68	1.09	0.05	0.13	0.73	1.05	0.81	1.35

Abbreviations: B, coefficient estimates; SE, standard error; LLCI, lower level of the 95% confidence interval; ULCI, upper level of the 95% confidence interval.

## Data Availability

The datasets generated during the current study are not publicly available but are available from the corresponding author on reasonable request.

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
