# Peer review of "Has the COVID-19 Pandemic Changed Parental Attitudes and Beliefs Regarding Vaccinating Their Children against the Flu?"

_vaccines, 2023, doi:10.3390/vaccines11101519_

Round 1
Reviewer 1 Report
The study is interesting because it evaluates the aptitute or hesitancy of a sample of parents toward vaccinating their children against the flu, following previous experience with Sars-Cov-2 vaccination during the pandemic. The scientific soundness is high.
The study design is appropriate but it could not be immediately understandable for readers with little knowledge of statistic. For instance, I understand the choise of Multinomial logistic regression analysis given the number of categotically dependent variables with two/or more possible outcomes (but also a Cox regression or a modified Poisson regression could have been used). Anyway, I think it opportune to add a legend to Table 3 of the main indicators, i.e. lower limit confidence interval (LLCI), upper limit confidence interval (ULCI), estimated multinomial logistic regression coefficient (B), etc. as not every reader may have sufficient knowledge of statistical methods for epidemiologic research.
Conclusion
In my opinion, the wheight of so called "cospiracy theories" has probably a very low influence in parent's decision. Whatever, it is not possible to evaluate it. Direct experience of pandemic management and social-media use could also be other reasons for hesitancy
Minor points:
Line 61, reference 9 is shown twice
Author Response
RE: Has the COVID-19 Pandemic Changed Parental Attitudes and Beliefs Regarding Vaccinating their Children against the Flu? vaccines-2583212
Reviewer #1: 25 Aug 2023
Comments and Suggestions for Authors
Comment 1. The study is interesting because it evaluates the aptitute or hesitancy of a sample of parents toward vaccinating their children against the flu, following previous experience with Sars-Cov-2 vaccination during the pandemic. The scientific soundness is high.
Answer 1: I would like to thank the reviewer for reviewing our manuscript and providing valuable feedback. I have carefully considered the comments and have made the required revisions to improve the manuscript.
Comment 2: The study design is appropriate but it could not be immediately understandable for readers with little knowledge of statistic. For instance, I understand the choise of Multinomial logistic regression analysis given the number of categotically dependent variables with two/or more possible outcomes (but also a Cox regression or a modified Poisson regression could have been used). Anyway, I think it opportune to add a legend to Table 3 of the main indicators, i.e. lower limit confidence interval (LLCI), upper limit confidence interval (ULCI), estimated multinomial logistic regression coefficient (B), etc. as not every reader may have sufficient knowledge of statistical methods for epidemiologic research.
Response 2: As per the first part of the reviewer’s comment, I added the following clarifying sentence to the Statistical Analyses sub-section: “Multinomial logistic regression is an appropriate statistical method due to the categorical nature of the dependent variable, ‘willingness to vaccinate’ measured in 3 distinct and non-ordinal categories: (1) Yes, I have already vaccinated my child; (2) I intend to vaccinate my child in the current winter, and (3) No, I have not vaccinated my child and do not intend to”. Regarding the second part of the comment, I added the legend to Table 3: “Abbreviations: B, coefficient estimates; SE, standard error; LLCI, lower level of the 95% confidence interval; ULCI, upper level of the 95% confidence interval”.
Conclusion
In my opinion, the wheight of so called "cospiracy theories" has probably a very low influence in parent's decision. Whatever, it is not possible to evaluate it. Direct experience of pandemic management and social-media use could also be other reasons for hesitancy
Comment 3. Minor points: Line 61, reference 9 is shown twice
Answer 3: Fixed.

Reviewer 2 Report
This manuscript offers an important exploration of parental attitudes and beliefs regarding vaccinating their children. There are several issues that I believe, if addressed, would result in a much stronger publication.
Introduction:
-Page 1: As per the NIH language guide, whenever possible, avoid labeling nations as developing, preferred language is low- and middle-income countries (LMIC).
-Page 2, line 76: consider replacing "explain" with "are associated with"
-Page 2, lines 76-78 - Given the focus on HBM, consider a short sentence explaining the factors you are most interested in
-Page 3, line 90, consider replacing "many children" with the exact percentage based on country epi data
-Given that the focus here is on Israeli parents, the introduction is too broad. More of a focus on the population of interest is warranted. Early in the COVID-19 pandemic, Israel led the world in vaccination uptake. An exploration of vaccine uptake and associated attitudes for Israeli people seems warranted here as you are exploring how these have changed since COVID. It is unclear why there is some focus on the US and other places without linking this to Israeli attitudes and data specific to Israeli populations.
Methods:
-Page 4, lines 110-114: Is this survey pre-test data published? If so, please cite.
-Page 4, line 131 - "intent" should be "intend"
Results:
-Page 6, lines 186-195 - Need to state upfront how many potential participants were approached to achieve the 975 sample size. Were there differences in those who did participate and those who did not? Could this be a limitation? Authors provide info in the supplementary materials, but this really needs to be included in the main text. If not differences still need to say so. Paragraph 2 of results section would be a good place to address this.
-Consider the use of the words "recalcitrant" and "recalcitrance" - this seems judgmental and potentially stigmatizing. It implies that participants have an obstinately uncooperative attitude toward authority or discipline. Your data have nothing that I see to support this.
Discussion:
-Do not repeat your results verbatim in the discussion. This section is meant to be a discussion of your results. Why they are important, how they contribute to the literature, adding to it, contradicting existing data, etc.
-Page 11, line 352: This would be a great place to discuss how your data might inform the use of HBM factors to develop interventions for your population. Have there been other interventions that worked? Might they be enhanced by including HBM factors? If so, what do your data point to in terms of what might be effective?
-Page 12, line 364: How exactly might increased awareness lead to improved vaccine uptake?
-Page 12, lines 370-373: Do you have any insight into what may have led to mistrust in the MOH? Any data to support shed light on this and how might HBM used to address this?
Study Limitations:
-As stated above, if there were any differences between those who participated in the survey and those who did not, need to state that as a potential limitation (as well as describing differences above as noted)
Conclusions:
-Page 13, lines 411-413: How does your data inform the way forward with regard to school-based vaccination programs? This could be part of your discussion.
A few minor spelling errors. Please review to ensure that there are no spelling or grammatical errors.
Author Response
RE: Has the COVID-19 Pandemic Changed Parental Attitudes and Beliefs Regarding Vaccinating their Children against the Flu? vaccines-2583212
Reviewer #2: 25 Aug 2023
Comments and Suggestions for Authors
This manuscript offers an important exploration of parental attitudes and beliefs regarding vaccinating their children. There are several issues that I believe, if addressed, would result in a much stronger publication.
Response: I would like to thank the reviewer for reviewing our manuscript and providing valuable feedback. I have carefully considered the comments and have made the required revisions to improve the manuscript.
Introduction:
Comment 1. Page 1: As per the NIH language guide, whenever possible, avoid labeling nations as developing, preferred language is low- and middle-income countries (LMIC).
Answer 1: I would like to thank the reviewer for this comment. The text was modified accordingly: “Previous studies have estimated that 99% of deaths in children under 5 years of age with lower respiratory tract infections associated with influenza were in low- and middle-income countries (LMIC) (Nair et al., 2011; WHO, 2023)”.
Comment 2. Page 2, line 76: consider replacing "explain" with "are associated with"
Answer 2: The text was modified accordingly: “This model proposes several factors that are associated with influenza vaccine acceptance….”.
Comment 3. Page 2, lines 76-78 - Given the focus on HBM, consider a short sentence explaining the factors you are most interested in
Answer 3: Following the reviewer’s comment, I added the following sentence to the introduction section: “Previous studies conducted prior to the COVID-19 pandemic have shown that the most dominant factors influencing parents' intentions to vaccinate based on the HBM are their perceived benefits, perceived susceptibility, and severity (Ben Natan et al., 2016; Chen et al., 2011; Malosh et al., 2014)”.
I also added the following sentence to the discussion section:
“Our study has illuminated the significance of HBM variables, particularly perceived susceptibility, perceived severity, and perceived benefits, which displayed the largest effect sizes in the context of vaccine acceptance. Our findings, which are in line with previous studies in the context of flu vaccination (Ben Natan et al., 2016; Chen et al., 2011; Malosh et al., 2014), underscore the critical role of these constructs as predictors of vaccine acceptance and their utility for interventions designed to promote health behaviors.”
Comment 4. Page 3, line 90, consider replacing "many children" with the exact percentage based on country epi data
Answer 4: Following the reviewer’s comment I added the exact percentage in Israel as follows:
“In 2021, over 50% of parents in Israel refused to vaccinate their children in these grades at school (Israeli Ministry of Health, 2022, pp. 2020–2021; Ministry of health, 2016)”.
Comment 5. Given that the focus here is on Israeli parents, the introduction is too broad. More of a focus on the population of interest is warranted. Early in the COVID-19 pandemic, Israel led the world in vaccination uptake. An exploration of vaccine uptake and associated attitudes for Israeli people seems warranted here as you are exploring how these have changed since COVID. It is unclear why there is some focus on the US and other places without linking this to Israeli attitudes and data specific to Israeli populations.
Answer 5: I would like to thank the reviewer for this valuable comment. Following it, I have added the following two paragraphs to the introduction section:
“In order to increase vaccination rates, since 2008, Israel has begun providing free influenza vaccinations for the entire population over the age of 6 months, and the Ministry of Health and the Health Maintenance Organizations (HMOs) have implemented various strategies to promote vaccination, especially among high-risk groups. Nonetheless, despite these efforts, studies conducted before the COVID-19 pandemic found a low degree of flu vaccine compliance in many countries across the globe, including Israel when it came to vaccination of children. Specifically, in the years 2010–2017, only 51–59% of this age group got vaccinated in the U.S. [4,5], and in Israel the numbers were even lower: 23-32% [6–8]. These figures are far below the Healthy People 2020 goal of 70% for children [9].
In the early stages of the COVID-19 pandemic, Israel emerged as a global leader in vaccination efforts, with remarkably high uptake rates of COVID-19 vaccine [10], compared to efforts for the flu vaccine. Following an extensive COVID-19 vaccine campaign, it is intriguing to examine the factors that will impact parents on their decision to vaccinate their children against influenza following the pandemic.”
Methods:
Comment 6. Page 4, lines 110-114: Is this survey pre-test data published? If so, please cite.
Answer 6: I would like to thank the reviewer for this comment. I revised the relevant text to clarify that the 100 respondents in the pre-test are also part of the larger cohort described in this manuscript:
“Before distributing the questionnaire to all of the respondents, a pre-test was conducted on the first 100 respondents to examine the reliability of the questionnaire. The reliability was verified using a Cronbach α internal reliability test (the HBM section of the questionnaire obtained an internal consistency of Cronbach α=0.85). These 100 respondents were also included in the larger study reported in this manuscript.”.
Comment 7. Page 4, line 131 - "intent" should be "intend"
Answer 7: Fixed.
Results:
Comment 8. Page 6, lines 186-195 - Need to state upfront how many potential participants were approached to achieve the 975 sample size. Were there differences in those who did participate and those who did not? Could this be a limitation?
Answer 8: I would like to thank the reviewer for this valuable comment. Following this comment, I have added the following footnote to the Results section:
“An invitation to fill out the questionnaire was sent to an online panel of 7,200 participants with the goal of obtaining ~1,000 filled questionnaires. The invitation to fill out the questionnaire expired after reaching 975 valid answers.”
I would like to emphasize that although I don’t have information on the panel participants who were exposed to the invitation but didn’t participate in the survey, those who did answer constitute a representative sample.
Authors provide info in the supplementary materials, but this really needs to be included in the main text. If not differences still need to say so. Paragraph 2 of results section would be a good place to address this.
Response: Following the reviewer's comment, I have revised the manuscript and provided more details from supplementary Table 2 in the main text as follows:
“The distribution of the sample closely matches that of the adult Jewish population in Israel (Supplementary Table 2). This is unsurprising as the sample was drawn in layers to satisfy certain characteristics (e.g., age, gender, level of religiosity, and geographical area). For example, approximately 60% of both the sample and the entire population were between the ages of 18 and 39.”
Comment 9. Consider the use of the words "recalcitrant" and "recalcitrance" - this seems judgmental and potentially stigmatizing. It implies that participants have an obstinately uncooperative attitude toward authority or discipline. Your data have nothing that I see to support this.
Answer 9: I completely agree and replaced the word recalcitrance with the word "unwillingness":
“Interestingly, parents whose children did not receive a COVID-19 vaccine in the year 2022 expressed significantly higher unwillingness to vaccinate their children against flu…”
Discussion:
Comment 10. Do not repeat your results verbatim in the discussion. This section is meant to be a discussion of your results. Why they are important, how they contribute to the literature, adding to it, contradicting existing data, etc.
Answer 10: Following this and the subsequent comments of the reviewer, I have revised the discussion section considerably. Specifically, I now elaborate more on why my findings are important, their contribution to the existing literature, etc. It should be noted that I still kept a very short paragraph at the beginning of the discussion section summarizing what was done in this study, to provide context for the rest of this section.
Comment 11. Page 11, line 352: This would be a great place to discuss how your data might inform the use of HBM factors to develop interventions for your population. Have there been other interventions that worked? Might they be enhanced by including HBM factors? If so, what do your data point to in terms of what might be effective?
Answer 11: Following the reviewer's valuable feedback, I have revised the corresponding paragraph in the discussion section as follows:
“Our study has illuminated the significance of HBM variables, particularly perceived susceptibility, perceived severity, and perceived benefits, which displayed the largest effect sizes in the context of vaccine acceptance. Our findings, which are in line with previous studies in the context of flu vaccination (Ben Natan et al., 2016; Chen et al., 2011; Malosh et al., 2014), underscore the critical role of these constructs as predictors of vaccine acceptance and their utility for interventions designed to promote health behaviors. To formulate effective interventions, it is important to take into account previous successful strategies. Prior research has suggested that increasing parental perception of benefits (such as reducing the risk of infection or protecting those at high risk), severity, and perceived susceptibility may be effective strategies for public health interventions. Therefore, directing public health messages towards parents that may increase perceptions of susceptibility and severity of influenza, and perceived benefits of the vaccine may yield more favorable outcomes in terms of increasing vaccine uptake [22].”
Comment 12. Page 12, line 364: How exactly might increased awareness lead to improved vaccine uptake?
Answer 12: Following the reviewer's valuable feedback, I have revised the relevant sentence in the discussion section as follow:
“The COVID-19 pandemic increased public awareness regarding the critical role vaccines play as the ultimate strategy for preventing infections and improving protection. It highlighted both human vulnerabilities to emerging infectious diseases, and the limitations of other disease control means (e.g., lockdowns). This awareness is likely to reflect in improved vaccine uptake in other diseases, including influenza [31]”.
Comment 13. Page 12, lines 370-373: Do you have any insight into what may have led to mistrust in the MOH? Any data to support shed light on this and how might HBM used to address this?
Answer 13: Following the reviewers' comment, I revised the following paragraph in the discussion section to shed some light on what may have led to mistrust in the MOH:
“Interestingly, lack of trust in the Ministry of Health and in the pharmaceutical companies increased among 78% of those unwilling to vaccinate their children in our study following the COVID-19 events. This lack of trust can arise from a range of factors, such as misinformation, lack of transparency and conspiracy theories about vaccines, all of which contribute to confusion and mistrust among some parents. Israel played a pioneering role as the first nation to embark on mass COVID-19 vaccination efforts. Prior to the release of early Israeli studies on the real-world effects of mass vaccination, considerable uncertainty prevailed regarding the vaccine's effectiveness beyond the controlled environment of clinical trials. Another factor contributing to this mistrust was the non-disclosure of certain details regarding the purchase agreement with Pfizer at the time (Rosen et al., 2021). It is, therefore, important that healthcare providers and public health officials address these concerns provide accurate information about the safety and efficacy of the influenza vaccine and work towards rebuilding trust in the Ministry of Health to help increase the vaccination rate among parents and children.”
Regarding the HBM model, it is important to emphasize that the HBM model was used in this study in the context of explaining/predicting the intention to vaccinate children and not with regard to the mistrust in the MOH, which was a separate independent addition to the questionnaire.
Study Limitations:
Comment 14. As stated above, if there were any differences between those who participated in the survey and those who did not, need to state that as a potential limitation (as well as describing differences above as noted)
Answer 14: Unfortunately, I don’t have information on the panel participants who were exposed to the invitation but didn’t participate in the survey. However, I would like to emphasize that those who did respond constitute a representative sample.
Following this comment, I have added the following limitation to the discussion section:
“Another limitation of this study is the lack of information regarding the panel participants who received invitations but did not participate in the survey. However, it should be noted that the panel participants who did respond constitute a representative sample.”
Conclusions:
Comment 15. Page 13, lines 411-413: How does your data inform the way forward with regard to school-based vaccination programs? This could be part of your discussion.
Answer 15: Following the reviewer's comment I have added the following paragraph to the discussion section:
“In Israel, influenza vaccination is not mandatory for school attendance, resulting in lower coverage rates compared to routine childhood vaccination. To increase vaccination uptake, several actions can be taken. First, nursing staff in student health centers should be made more accessible. Second, clearer explanations should be provided about the effectiveness of nasal spray vaccinations. Finally, more efforts should be invested in simple outreach activities (Glatman-Freedman et al., 2019). For example, a recent research has shown that sending behaviorally informed invitation letters and reminders can increase childhood influenza vaccinations through schools and general practitioners (GP) (Howell-Jones et al., 2023)”.

Round 2
Reviewer 2 Report
The increased focus on Israeli data is a great improvement as is the brief overview of the HBM. The authors were incredibly responsive to the issues brought up in the review and this is a much improved manuscript. Really great data.